# Expedient production of site specifically nucleobase-labelled or hypermodified RNA with engineered thermophilic DNA polymerases

Mária Brunderová [1,2,3], Vojtěch Havlíček[1,2], Ján Matyašovský[1], Radek Pohl [1], Lenka Poštová Slavětínská[1], Matouš Krömer [1,4] ✉ & Michal Hocek [1,2] ✉

Innovative approaches to controlled nucleobase-modified RNA synthesis are urgently needed to support RNA biology exploration and to synthesize potential RNA therapeutics. Here we present a strategy for enzymatic construction of nucleobase-modified RNA based on primer-dependent engineered thermophilic DNA polymerases – SFM4-3 and TGK. We demonstrate introduction of one or several different base-modified nucleotides in one strand including hypermodified RNA containing all four modified nucleotides bearing four different substituents, as well as strategy for primer segment removal. We also show facile site-specific or segmented introduction of fluorophores or other functional groups at defined positions in variety of RNA molecules, including structured or long mRNA. Intriguing translation efficacy of single-site modified mRNAs underscores the necessity to study isolated modifications placed at designer positions to disentangle their biological effects and enable development of improved mRNA therapeutics. Our toolbox paves the way for more precise dissecting RNA structures and functions, as well as for construction of diverse types of base-functionalized RNA for therapeutic applications and diagnostics.

RNA is fundamental molecule of life with plethora of biological roles and mechanisms. Its importance is given not only by its coding and structural potential, but also due to extensive chemical diversity. Nucleoside modifications are ubiquitous throughout all classes of RNA molecules. More than 170 natural modifications have been discovered in RNA so far and this number is constantly expanding[1]. RNA modifications modulate wide range of cellular functions[2,3] and their aberrations play pivotal role in disease development (i.e., cancer[4], metabolic[5] and neurological disorders[6] and cardiovascular conditions[7]). Despite growing understanding of physiological and

pathological impact, many modifications remain orphans–with uncharacterized mechanisms of writing, reading and unknown functional consequences[8].

Appreciation of biological importance of RNA gave rise to a variety of RNA-based therapies, that enabled targeting previously deemed "undruggable" processes[9–11]. Expression of therapeutic genes of interest can be elicited by intracellular delivery of mRNA[12]. Remarkable efficacy of SARS-CoV-2 mRNA vaccines spurred rapid development in the field[13,14]. Most successful therapeutic mRNA molecules rely on extensive nucleobase modifications that can improve stability,

[1]Institute of Organic Chemistry and Biochemistry, Czech Academy of Sciences, Flemingovo nam. 2, CZ-16000, Prague 6, Czech Republic. [2]Department of Organic Chemistry, Faculty of Science, Charles University, Hlavova 8, CZ-12843, Prague 2, Czech Republic. [3]Present address: MRC Laboratory of Molecular Biology, Francis Crick Avenue, Cambridge Biomedical Campus, Cambridge, UK. [4]Present address: The Rosalind Franklin Institute, Harwell Campus, Didcot, Oxfordshire, UK. ✉e-mail: matous.kromer@gmail.com; hocek@uochb.cas.cz

**Fig. 1 | Comparison of currently used methods with the presented engineered DNA polymerase-based route to base-modified RNA.** Created with BioRender.com.

increase the translational efficiency and modulate other properties[15,16]. A case in point represents N1-methylpseudouridine (m1Ψ), that shields RNA from innate immunity receptors, alters mRNA secondary structure, and leads to direct enhancement of mRNA translation[17,18]. Notably, the chemical space of other non-natural mRNA modifications has not been explored systematically yet.

Dissecting RNA structure, dynamics, function, and deciphering effects of both natural and designer chemical modifications, as well as development of emerging RNA therapies, warrant access to synthetic modified RNA with suitable functional groups (e.g., epitranscriptomic tags, fluorescent, spin or isotopic labels)[19,20]. Nevertheless, contemporary methods of solid-phase chemical synthesis allow to reliably obtain oligonucleotides only up to 50–60 nucleotides long, which fall short of the length of many biologically relevant RNAs, harsh conditions can jeopardize modifications integrity[21], building blocks are unstable and the process is not environmentally friendly[22]. Greener biocatalytic production of oligonucleotides has been recently developed, however, no nucleobase modifications have been incorporated and this method is limited to shorter fragments [<25-nucleotide (nt)][23]. Construction of larger modified RNAs by stepwise enzymatic ligation necessitates intricate design of fragments, might be hindered due to structural constraints, suffers from laborious purification and poor yields[24,25]. Site selective modifications can be installed with assistance of substrate promiscuous enzymes, e.g., methyltransferases, trans-glycosylases or ribozymes[26]. This route is restricted to terminal labeling or to predetermined consensus motifs[27].

Enzymatic in vitro transcription (IVT) mediated by T7 RNA polymerase (T7 RNAP)[28] allows to attain longer sequences[29]. We and others have extensively explored landscape of nucleobase-modified RNA synthesis by IVT with T7 RNAP[30–35]. It has been shown that this enzyme has limited tolerance to bulky or 7-deazaguanine modifications[32], requires optimal GGG trinucleotide initiation sequence[36] and is prone to termination when transcribing heavily structured RNA[37]. In addition, inclusion of base-modified ribonucleoside triphosphates (rNXTPs)[38] in IVT typically renders RNA uniformly modified at all positions[30,34,39] (Fig. 1). Notable exception is position-selective labeling by T7 RNAP (PLOR)[33], whereby omitting one nucleotide from IVT allows to pause synthesis at missing nucleotide site and to incorporate modification at this position (Fig. 1). The method, however, requires extensive optimization of reaction parameters.

On the other hand, we and others have repeatedly demonstrated thermostable DNA polymerases to be highly efficient enzymes in construction of even heavily modified DNA from base-modified 2′-deoxyribonucleoside triphosphates (dNXTPs)[40,41]. However, DNA polymerases are not amenable to incorporation of ribonucleoside triphosphates (rNTPs) to avoid disruption of genome integrity[42]. This natural constraint was recently unlocked with power of directed evolution. The Romesberg group evolved SFM4-3 polymerase, derived from Stoffel fragment of *Thermococcus aquaticus* (Taq) A-family polymerase by phage display focused on incorporation of C2′-modified DNA[43,44]. Simultaneously, the Holliger group engineered TGK polymerase, derived from parental *Thermococcus gorgonarius* B-family DNA polymerase (Tgo) by mutating two key steric gatekeeping amino acids[45]. Both enzymes were proficient in synthesis of natural RNA or xenonucleic acids (XNA) with 2′-sugar modification, however, none of them has been examined for polymerization of nucleobase-modified rNXTPs leading to base-modified RNA, that may find diverse attractive applications in molecular biology, therapy, or diagnostics.

Here we disclose a systematic study (Fig. 1) exploring the scope of engineered DNA polymerases for incorporation of manifold rNXTPs, in various combinations and sequence contexts and comparison of these methods to the conventional IVT with T7 RNAP. We further investigated applications in site-specific labeling of RNA for functional and dynamic studies or biotechnology applications.

## Results

### Engineered polymerases are superior to T7 RNAP for base-modified RNA construction

To delineate the scope of modifications accepted by engineered mutant DNA polymerases (Supplementary Information Section 2.4.1, 2.4.2) and to study the impact of modification characteristics on polymerase performance in primer extension reaction (PEX), we designed a library of modified rNXTPs (Fig. 2, Supplementary Figs. S2–S5). Modifications were attached to the position 7- of 7-deazapurine or to position 5- of pyrimidine bases, which makes them well tolerated by cognate DNA polymerases[46]. We synthesized a full set of unnatural rNXTPs bearing clickable (ethynyl, E) or hydrophobic groups of increasing bulkiness, pentyn-1-yl (Pent) and phenyl (Ph), attached to each pyrimidine or 7-deazapurine nucleotide (rAXTP, rUXTP, rCXTP, rGXTP). Further we installed reactive functionalities

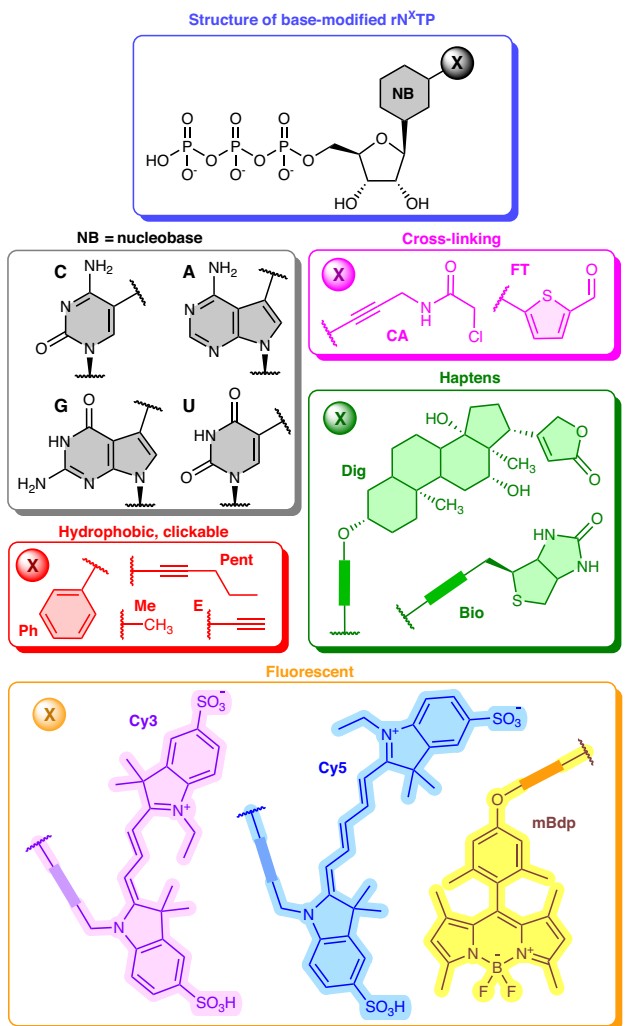

**Fig. 2 | Structures of base-modified nucleotides (rN^XTPs) used in this study.** For full chemical structures of the rN^XTPs see Supplementary Figs. S2–S5. For synthesis of E-, Pent-, Ph-, mBdp-, CA- and FT-modified rN^XTPs see Supplementary Information Section 1 (Synthetic part), for NMR characterization see Supplementary Figs. S89–S125.

chloroacetamide (CA) or formylthienyl (FT) on rA^XTP, rU^XTP and rC^XTP, which are useful for covalent capturing of RNA interacting proteins[35]. We also synthesized rC^mBdpTP bearing fluorescent tag (mBdp) and also included commercial rN^XTPs bearing cyanine dyes (Cy5, Cy3) and affinity tags (Bio, Dig), as well as epitranscriptomic (Me) nucleobase modifications.

Initially, we tested incorporation of modified rN^XTPs in PEX requiring insertion of one modified nucleotide at single internal position, followed by three unmodified rNTPs (Fig. 3Aa) with usage of fluorescently labeled RNA primers – FAM-RNA-prim_15nt or Cy5-RNA-prim_15nt (Supplementary Table S3) and ssDNA template – templ_19_nt_X (X = A, U, C, G) (Supplementary Table S5). PEX were carried out at optimal temperature 60 °C. In each case of the tested rN^XTPs we observed formation of full-length products on denaturing PAGE (dPAGE) confirming that the mutant polymerases are capable to incorporate a variety of modified substrates (Fig. 3Ca, Supplementary Figs. S6–S30). Single-stranded modified RNA products were obtained from double-stranded DNA-RNA hybrids by mild template degradation using TurboDNase and confirmed by MS-MALDI (Supplementary Figs. S126–S180). For CA-bearing rA^CATP and rC^CATP, where side conjugation reactions were detected by MS (Supplementary Figs. S129,

S130, S147, S148, S158, S159, S174, S175, S211, S212, S261, and S262), the temperature was reduced to 37 °C or reaction times were shortened, which mitigated formation of intramolecularly cyclised by-products (Supplementary Figs. S132, S133, S150, S151, S161, and S177). Further we investigated multiple incorporations (Fig. 3Ab) of each modified nucleotide using FAM-RNA-prim_15nt or Cy5-RNA-prim_15nt and 5´-(TINA)-templ_31nt or 5´-(dual-Bio)-templ_31nt (Supplementary Table S5) encoding for 31-nt long RNA with four modification sites. Neither TGK (Fig. 3Cb, Supplementary Figs. S31–S37) nor SFM4-3 (Supplementary Figs. S38–S43) polymerase have displayed any difficulties in PEX with more challenging DNA template giving clean full-length products, whose identities were confirmed either by MS-MALDI (Supplementary Figs. S181–S210) or LC-ESI-MS (Supplementary Figs. S213–S216).

To benchmark engineered polymerases against T7 RNAP we designed challenging templ_50nt (Supplementary Table S5) for PEX of Cy5-RNA-prim_15nt or dsDNA template (ds-templ_52nt, Supplementary Table S6) for IVT where the synthesized part of RNA is identical (Fig. 3Ac, Ad) and tested simultaneous incorporation of several different modified nucleotides. We used either a mixture of three phenyl-modified (rA^PhTP, rU^PhTP, rC^PhTP, Mix-1), three distinct modified (rA^ETP, rU^BioTP, rC^PhTP, Mix-2) or sterically demanding (rA^PhTP, rU^BioTP, rC^mBdpTP, Mix-3) nucleotides in combination with natural rGTP. As it was expected, the T7 RNAP substantially failed in synthesis of modified RNA with most modifications (Mix-1, Mix-3), while some full-length product was observed for Mix-2 (Supplementary Fig. S63). SFM4-3 synthesized RNA with Mix-1 and Mix-2 with only modest yield and entirely did not accept Mix-3 (Supplementary Fig. S61). On the other hand, TGK polymerase was successful in accepting all rN^XTP mixtures giving high yield of full-length products in each case (Supplementary Fig. S60). Additionally, the efficiency of the mutant polymerases was compared to T7 RNAP in synthesis of homopolymeric termination-prone poly-U sequence (Fig. 3Af, Ag) encoding for a stretch of modified rU^Bio nucleotides with usage of either ssDNA templ_poly-U (Supplementary Table S5) and FAM-RNA-prim_15nt in PEX or dsDNA template – ds-templ_poly-U (Supplementary Table S6) for IVT. TGK polymerase delivered significant amount of modified ssRNA (Fig. 3Cf, Supplementary Fig. S59), whereas both SFM4-3 (Supplementary Fig. S59) and T7 RNAP (Fig. 3Cg, Supplementary Fig. S62) did not succeed in synthesis of neither natural nor modified RNA. To quantify the activity of engineered DNA polymerases, we conducted simple kinetic studies (Fig. 3B, Supplementary Figs. S57 and S58) using 15-nt RNA (Cy5-RNA-prim_15nt) or DNA primer (Cy5-DNA-prim_15nt, Supplementary Table S4) and 16-nt DNA template (templ_16nt, Supplementary Table S5) for single-nucleotide-incorporation (SNI) of rC^mBdpTP at different time periods. TGK polymerase provided higher conversion than SFM4-3 for both primer types in virtually all time points. Analogously, we monitored SNI of rC^mBdpTP with varying concentrations of SFM4-3 or TGK polymerase at fixed time, which further supported higher activity of TGK (Supplementary Figs. S55 and S56).

In certain cases, the negative control experiments performed in absence of one natural rNTP showed some formation of spurious full-length product (e.g. Supplementary Figs. S12–S16) presumably due to misincorporation. For this reason, we have further tested the polymerase fidelity for incorporation of modified rN^XTPs. We performed single nucleotide extension of Cy5-RNA-prim_15nt complementary to templ_16nt encoding for one rC nucleotide with exceptionally bulky rC^mBdpTP mixed at different ratios with three natural nucleotides (rATP, rUTP, rGTP) to determine misincorporation ratios. The use of this bulky nucleotide enabled clear distinction of the modified (correct product) and non-modified RNA (faulty product) on dPAGE. Gratifyingly, even with 6 equiv. excess of natural nucleotides (rATP, rUTP, rGTP) towards rC^mBdpTP, more than 90% of the product contained correct rC^mBdp nucleotide as determined by gel analysis, confirming

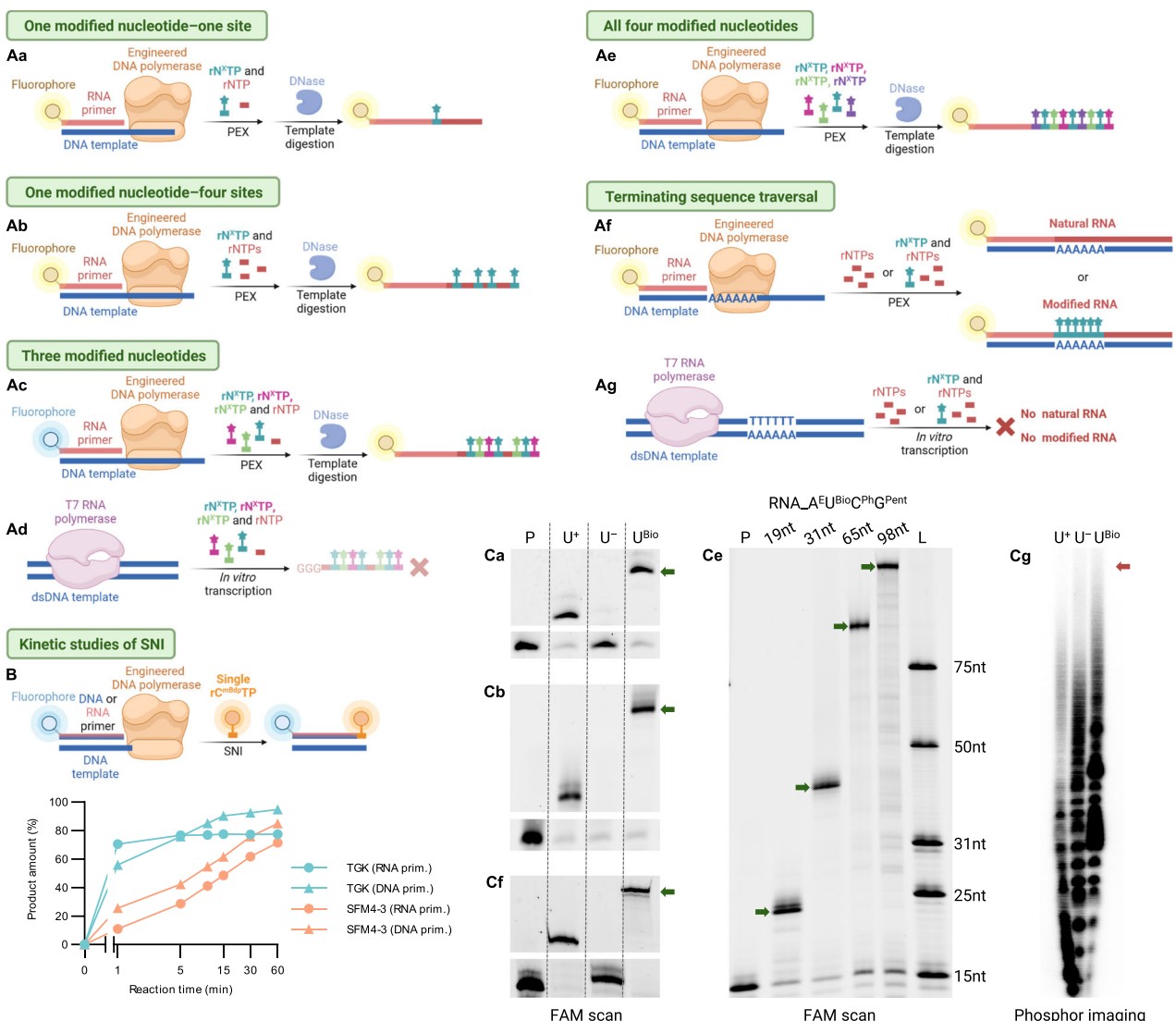

**Fig. 3 | Activity profiling of engineered thermostable DNA polymerases and T7 RNAP.** Templates requiring incorporation of single modification type at one (**Aa**, **Ca** dPAGE of PEX with TGK) or four (**Ab**, **Cb** dPAGE of PEX with TGK) sites were used to assess SFM4-3 and TGK polymerases when accepting modified rN^XTPs. T7 RNAP (**Ad**) and engineered polymerases (**Ac**) were compared using templates encoding for three types of modifications. Comparison of TGK and SFM4-3 polymerase was also performed in PEX with four rN^XTPs (rA^ETP, rU^BioTP, rC^PhTP, rG^PentTP) (**Ae**, **Ce** dPAGE of PEX with TGK and variously long templates). Eventually, by-pass of loosely hybridizing poly(A) sequence challenge was examined for engineered DNA polymerases (**Af**, **Cf** dPAGE of PEX with TGK) and T7 RNAP (**Ag**, **Cg** dPAGE of IVT with T7 RNAP). **B** Time course insertion of single nucleotide was monitored for SFM4-3 and TGK polymerase as percentage of primer conversion to product. TGK proved more active in all cases. Similar, consistent results have been obtained once for RNA primer, whereas experiment with DNA primer was only performed once. **Ca**, **Cb**, **Cf**, **Cg** Gels legend: (P) RNA primer; (U^+) positive control, PEX/IVT with natural rNTPs; (U^−) negative control, natural rNTPs without rUTP; (U^Bio) modification, PEX/IVT with a mixture of natural rNTPs where rUTP is replaced by rU^BioTP. For uncropped gel scans see Supplementary Figs. S10, S33, S59, and S62. **Ce** Gels legend: (19nt) PEX with templ_19nt_mix and incorporation of 4X rN^XTPs. (31nt) PEX with 5´-(TINA)-templ_31nt and incorporation of 16X rN^XTPs. (65nt) PEX with templ_65nt and incorporation of 50X rN^XTPs. (98nt) PEX with templ_98nt and incorporation of 83X rN^XTPs. (L) RNA ladder composed of FAM-labeled RNA oligonucleotides of indicated length. For uncropped gel scan see Supplementary Fig. S49. Final products are denoted by green arrow. No product formation is indicated by red arrow. Created with BioRender.com. Source data are provided as a Source Data File.

sufficient fidelity of the synthesis of base-modified RNA (Supplementary Figs. S64 and S65).

Since T7 RNAP can generate multiple copies of RNA from single DNA molecule, we were also interested in possibility of template recycling for PEX. We repeatedly denatured, annealed, and extended primer Cy5-RNA-prim_15nt with ssDNA template templ_98nt (Supplementary Table S5) and either natural rNTPs or a mixture of rATP, rUTP, rC^MeTP, rGTP. As a control, non-cycled reaction with single extension was carried out. Gel analysis revealed that cycling generates significantly higher amount of full-length product, however, also competing degradation of RNA was detected (Supplementary Fig. S88). Different strategy was needed for longer RNA due to their higher propensity to decay. The 1163-nt long dual-biotinylated ssDNA template – templ_(dual-Bio)-IRES-nLuc (Supplementary Table S5, Supplementary Fig. S78) immobilized onto streptavidin magnetic beads was successfully reused up to 5 times in solid-phase PEX with 23-nt long fluorescent RNA primer – Cy5-mRNA-prim (Supplementary Table S3) and either natural rNTPs or a mixture of rATP, rUTP, rC^MeTP, rGTP. The RNA products mRNA-nat or mRNA-full were liberated with mild NaOH elution after each PEX cycle, albeit with gradually decreasing yields as determined by agarose gel electrophoresis (Supplementary Figs. S86 and S87).

Next, we tested the ability of engineered polymerases to incorporate a mixture of all four different base-modified rN^XTPs (rA^ETP,

rU$^{Bio}$TP, rC$^{Ph}$TP, rG$^{Pent}$TP) in PEX (Fig. 3Ae) with variously long ssDNA templates (19-, 31-, 65- or 98-nt, Supplementary Table S5) encoding for incorporation of up to 83 consecutive modified nucleotides displaying four different functional groups. TGK polymerase provided full-length hypermodified RNA in all cases (Fig. 3Ce, Supplementary Figs. S45–S49) and all products were confirmed by ESI-MS (Supplementary Figs. S217–S224). On the other hand, SFM4-3 polymerase was inefficient already in synthesis of shorter hypermodified products (Supplementary Figs. S50–S53).

To probe length limitations of TGK-mediated synthesis of hypermodified RNA, we conducted PEX with either natural rNTPs or a mix of rA$^{E}$TP, rU$^{E}$TP, rC$^{Me}$TP, rG$^{Pent}$TP on ssDNA templates – templ_IRES, templ_IRES-prolonged, templ_IRES-nLuc (Supplementary Table S5, Supplementary Figs. S75–S77) varying in length (encoding for 606, 804 or 1147-nt long RNAs) and Cy5-mRNA-prim. Agarose electrophoresis revealed that templ_IRES and templ_IRES-prolonged produced full-length products with both natural and modified nucleotides (Supplementary Fig. S80). The longest template – templ_IRES-nLuc gave sharp band of full-length product in case of natural rNTPs, but only smeared band was observed for modified rN$^{X}$TPs (Supplementary Fig. S80). To confirm formation of products of correct length, we carried out reverse transcription and subsequent qPCR analysis to ensure that only cDNA (and not DNA template) is amplified (Supplementary Figs. S82, S84, and S85). Bands of expected lengths were obtained for all three modified RNA templates (Supplementary Figs. S81 and S83).

For certain applications, larger amounts of RNA probes might be needed. To test the methodology, we performed PEX on 1 nmol scale with TGK polymerase, templ_31nt (Supplementary Table S5) and FAM-RNA-prim_15nt in two replicates. Natural rNTPs or single modified nucleotide-containing mixtures (rA$^{E}$TP, rUTP, rCTP, rGTP); (rATP, rU$^{Bio}$TP, rCTP, rGTP); (rATP, rUTP, rC$^{Me}$TP, rGTP) or (rATP, rUTP, rCTP, rG$^{Pent}$TP) were used as representative examples of modifications on each nucleobase. After silica column purification, full-length products confirmed by MS (Supplementary Figs. S241–S260) were quantified against serially diluted synthetic standard by dPAGE (Supplementary Fig. S44). Yields of purified RNA varied from 63% to 78% depending on modification type (Supplementary Table S8).

## All-nucleobase modified RNA is generated through primer degradation

Since the performance of TGK polymerase generally surpassed SFM4-3, we decided to further proceed only with TGK polymerase. Although we proved that elongation of RNA primer is possible with a mixture of all four modified rN$^{X}$TPs, enzymatic generation of hypermodified RNA oligonucleotides, entirely modified at all four nucleobases has remained unaddressed challenge. Our first attempt was to generate hypermodified RNA with usage of DNA primer and template in PEX with rN$^{X}$TPs, followed by degradation of DNA segments with TurboDNase. We carried out PEX with FAM-labeled ssDNA primer (FAM-DNA-prim_15nt, Supplementary Table S4) complementary to 36-nt long ssDNA template (templ_36nt, Supplementary Table S5) and mix of rATP, rUTP, rC$^{Cy5}$TP, rGTP to visualize the RNA after DNA primer removal. Unfortunately, the usage of TurboDNase for DNA primer degradation led to multiple products, even with high DNase concentration or prolonged reaction time (Supplementary Fig. S66). Therefore, we slightly modified this protocol inspired by naturally occurring repair processes with uracil-DNA glycosylase (UDG)[47]. In this approach we used ssDNA primer containing single dU modification (FAM-DNA-prim_dU, Supplementary Table S4) for PEX with a mixture of rATP, rUTP, rC$^{Cy5}$TP, rGTP. Generated PEX products were treated in the first step by UDG to form abasic site, followed by cleavage under mild basic conditions using N,N'-dimethylethylenediamine (DMEDA) in the second step[48] and final DNA template (templ_36nt) degradation with TurboDNase (Supplementary Figs. S67, S225, and S226). This

workflow was then successfully exploited for generation of hypermodified RNA polymers prepared by PEX (Supplementary Fig. S68) with all four base-modified rN$^{X}$TPs (rA$^{E}$TP, rU$^{Bio}$TP, rC$^{Cy5}$TP, rG$^{Pent}$TP) (Fig.4A) and the desired final all-base modified RNA cleavage product was confirmed by dPAGE (Fig. 4B, Supplementary Fig. S69) and ESI-MS (Fig. 4C, Supplementary Figs. S227 and S228). To obtain fluorescent terminally labeled hypermodified RNA, we used primer carrying dU immediately followed by fluorescein-linked dT nucleotide (Cy5-DNA-prim_dU_dT-FAM, Supplementary Table S4), which is retained upon primer and template (templ_35nt, Supplementary Table S5) degradation, and allows RNA tracking (Fig. 4D). Thereby, we yielded 5'-terminally labeled fluorescent hypermodified RNA polymers build exclusively from modified rN$^{X}$TPs (rA$^{E}$TP, rC$^{Ph}$TP, rU$^{Bio}$TP, rG$^{Pent}$TP) confirmed by dPAGE (Fig. 4E, Supplementary Fig. S71) and ESI-MS (Fig. 4F, Supplementary Figs. S229 and S230).

## Site-specific labeling of RNA can be achieved by one-pot enzymatic cascade

We reasoned that our ability to precisely manipulate PEX conditions, in analogy to our prior study on DNA[49], would enable site-specific insertion of modifications at predefined positions also in RNA. We decided to modify the well-studied aptamer domain of the adenine riboswitch at specific positions of loop-1 and loop-2 with fluorescent tags to study the structure and dynamics using Förster (fluorescence) resonance energy transfer (FRET) measurements[50]. The enzymatic synthesis of FAM-Cy5-Cy3-triple-labeled riboswitch (FAM-Cy5-Cy3-riboswitch, Fig. 5A) was performed with combination of SNI and PEX using FAM-labeled RNA primer (FAM-RNA-prim_23nt, Supplementary Table S3) and two differently long ssDNA templates (templ_ribosw71_A, templ_ribosw71_B, Supplementary Table S5). Initially, RNA primer complementary to the first DNA template (templ_ribosw71_A) was extended by only one rU$^{Cy5}$ nucleotide by SNI. Degradation of the modified rU$^{Cy5}$TP by shrimp alkaline phosphatase (rSAP), followed by enzyme denaturation and addition of abundant amount of four natural rNTPs used in the second step led to complete elongation of the primer along DNA template (templ_ribosw71_A). Degradation of natural rNTPs was ensured by rSAP with concurrent template degradation using TurboDNase followed by heat denaturation of both enzymes. Addition of second, longer DNA template (templ_ribosw71_B) in presence of rU$^{Cy3}$TP led to second SNI and subsequent extension of the primer with large excess of natural rNTPs produced the full-length 71-nt long FAM-Cy5-Cy3-riboswitch (Fig. 5A). The progress of whole cascade was confirmed by dPAGE (Fig. 5B, Supplementary Fig. S72) and ESI-MS of each intermediate and final full-length product (Supplementary Figs. S231–S238). For conformational studies, we similarly prepared the Cy5-Cy3-labeled version of the riboswitch (Supplementary Figs. S239 and S240) and performed simple FRET measurements in presence of Mg$^{2+}$ and urea[51]. The prepared FRET probe exhibited (Fig. 5C), upon addition of adenine ligand, increased emission at 670 nm, when irradiated at 530 nm, while decreased emissions at 570 nm (Fig. 5D, Supplementary Fig. S74). This indicates conformational changes in riboswitch arms tertiary structure to occur in concordance with previous observations[50].

## Single-site modified mRNA leads to enhanced translation

Nucleobase modifications were game changers in the mRNA vaccines development[17]. Established methods for manufacturing mRNA vaccines harness T7 RNAP IVT, whereby modified nucleotide replaces natural congener[52]. This leads to substitution at all positions, which does not allow to discern sequence- and region-dependent impacts of nucleobase modification. Hence, we sought to apply our method to generate single-site modified mRNA and evaluated its performance in in vitro and in cellulo translation systems.

We designed an IRES-promoted[53] cap-independent RNA constructs (Fig. 6) coding for nanoluciferase[54], which produces

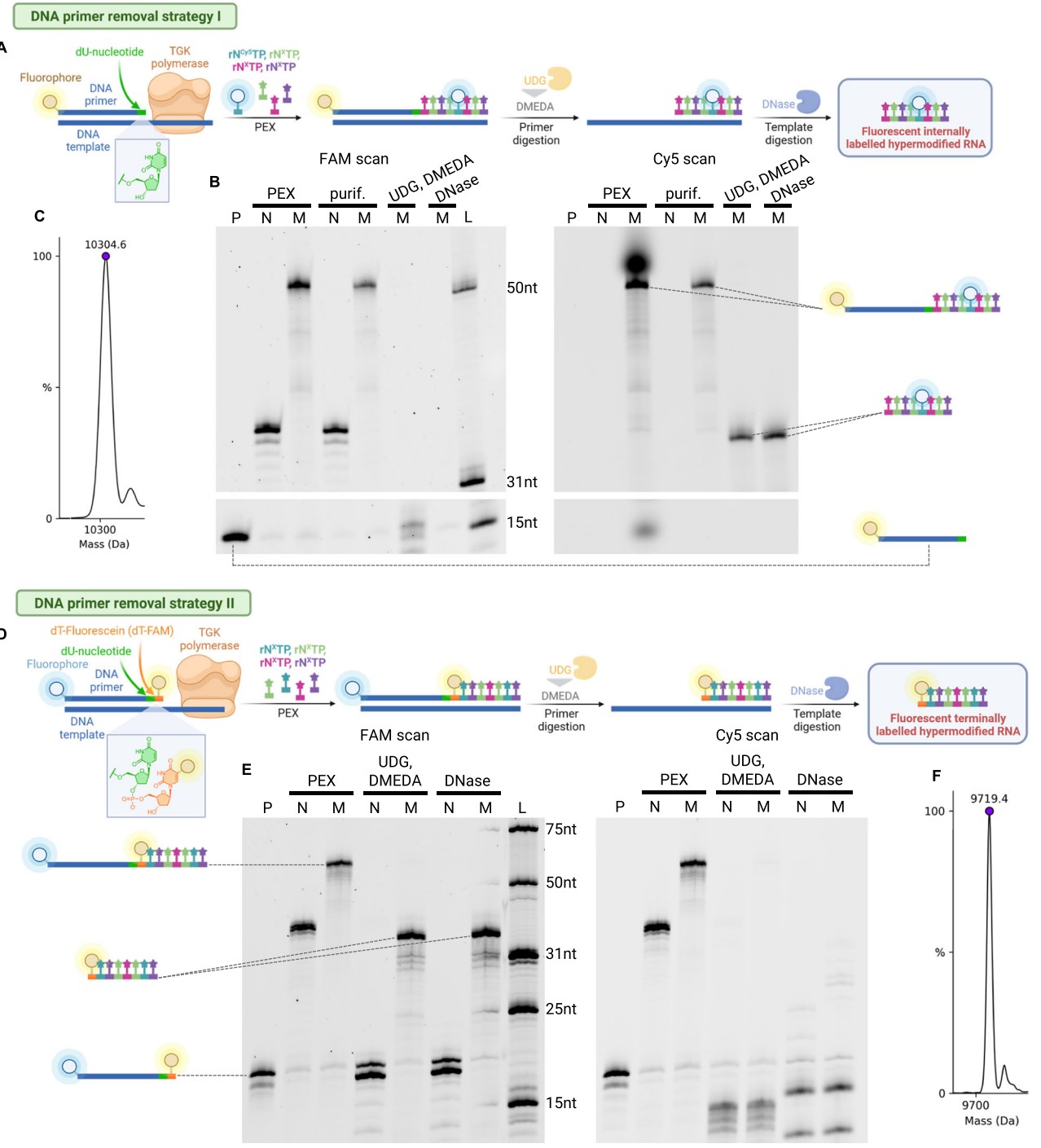

**Fig. 4 | DNA primer removal strategies. A** 5′-FAM labeled primer contains single 2′-dU modification at 3′-end terminus. Upon PEX with four rN$^X$TPs, one of them labeled with Cy5, dU is cleaved by UDG to generate abasic site, that is subsequently scissed by DMEDA under mild basic conditions. DNA template is degraded with DNase to produce fully modified RNA. **B** FAM and Cy5 scan of dPAGE of the aforementioned process. Cy5 scan allows to monitor RNA part of the molecule, while FAM scan is tracking fate of the DNA component. Gel legend: (P) DNA primer; (N) positive control, PEX with natural rNTPs; (M) modified RNA, PEX with a mixture of rA$^E$TP, rU$^{Bio}$TP, rC$^{Cy5}$TP, rG$^{Pent}$TP. (L) Ladder composed of FAM-labeled RNA oligo-nucleotides of indicated length. For uncropped gel scans see Supplementary Fig. S69. This experiment was repeated once with similar, consistent results. **C** Excerpt from ESI-MS deconvoluted spectrum of fully modified RNA. Calculated mass 10305.3 Da, measured mass 10304.6 Da. For full ESI-MS spectra see Supplementary Figs. S227 and S228. **D** 5′-Cy5 labeled primer contains single 2′-dU followed

by fluorescein tagged 2′-dT at 3′-end terminus. Upon PEX with four rN$^X$TPs, the primer and template are removed as described in **A**, however herein 5′-(fluorescein-dT) label is retained at RNA part allowing its detection. **E** FAM and Cy5 scan of dPAGE of the aforementioned process. FAM scan allows to monitor RNA part of the molecule, while Cy5 scan is tracking fate of the DNA component. Gel legend: (P) DNA primer; (N) positive control, PEX with natural rNTPs; (M) modified RNA, PEX with a mixture of rA$^E$TP, rU$^{Bio}$TP, rC$^{Ph}$TP, rG$^{Pent}$TP. (L) Ladder composed of FAM-labeled RNA oligonucleotides of indicated length. For uncropped gel scans see Supplementary Fig. S71. This experiment was repeated once with similar, consistent results. **F** Excerpt from ESI-MS deconvoluted spectrum of fully modified RNA. Calculated mass 9719.4 Da, measured mass 9719.4 Da. For full ESI-MS spectra see Supplementary Figs. S229 and S230. Created with BioRender.com. Source data are provided as a Source Data File.

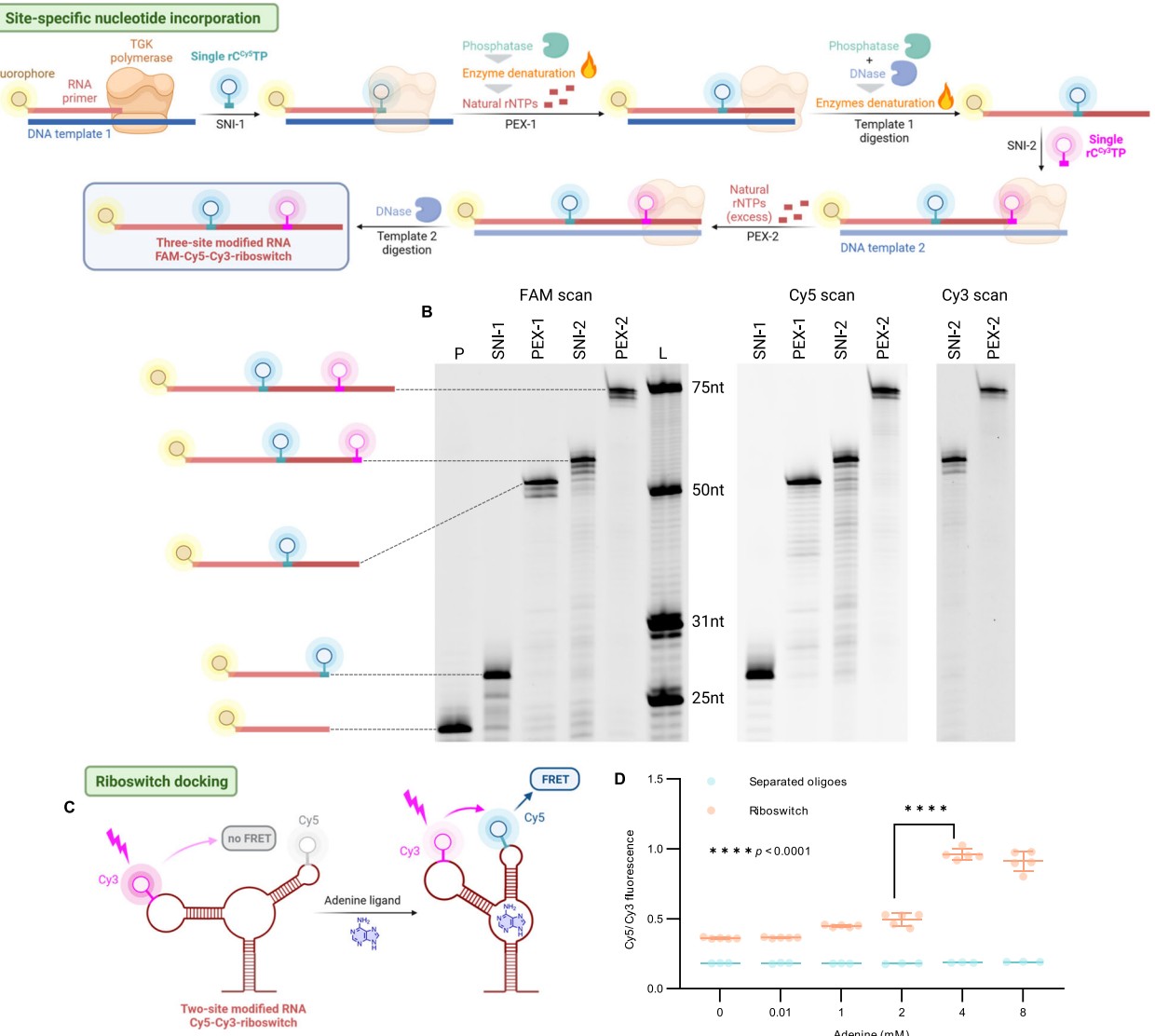

**Fig. 5 | Overall strategy of single-tube site-specific RNA labeling at two distinct positions. A** In the first step, the FAM-labeled RNA primer is prolonged by SNI of rC$^{Cy5}$TP. The unincorporated rC$^{Cy5}$TP is digested by phosphatase. The FAM-Cy5-labeled RNA is prolonged by PEX with a mixture of all four natural rNTPs. The DNA template 1 is removed with DNase. In the following step, the FAM-Cy5-RNA intermediate is used in second SNI with rC$^{Cy3}$TP and DNA template 2. Upon adding large excess of all four natural rNTPs, the desired full-length oligonucleotide is synthesized. DNA template 2 is finally degraded. **B** FAM, Cy5 and Cy3 scan of dPAGE of FAM-Cy5-Cy3-riboswitch. For uncropped gel scans see Supplementary Fig. S72. Gel legend: (P) RNA primer; (SNI-1) SNI of rC$^{Cy5}$TP; (PEX-1) PEX with natural rNTPs; (SNI-2) SNI of rC$^{Cy3}$TP; (PEX-2) PEX with natural rNTPs. (L) Ladder composed of FAM-labeled RNA oligonucleotides of indicated length. This experiment was repeated twice with similar, consistent results. **C** Two, Cy5 and Cy3 labeled, arms of riboA71 aptaswitch (Cy5-Cy3-riboswitch) are brought together upon adenine docking. Induced tertiary structural changes lead to augmented FRET signal. **D** Normalized FRET signals measured at various adenine concentrations. Two-way ANOVA test was used for statistical significance determination, error bars represent standard deviation from mean (FRET measurements were carried out in independent replicates, $n = 5$ for riboswitch, $n = 3$ for negative controls, $p$-value < 0.0001). For extended graph see Supplementary Fig. S74. Created with BioRender.com. Source data are provided as a Source Data File.

luminescence upon maturation. Single stranded DNA template was prepared from source plasmid (Supplementary Fig. S265) by PCR with subsequent strand separation on streptavidin magnetic beads. We opted to incorporate 5-methylcytidine nucleoside triphosphate (rC$^{Me}$TP), which was shown to be beneficial for mRNA translation performance[18]. First, we prepared control RNAs containing only unmodified nucleotides (Fig. 7A-1) or rC$^{Me}$ in whole length (Fig. 7A-2). The PEX was conducted with generated ssDNA template (templ_IRES-nLuc, Supplementary Table S5, Supplementary Fig. S77) and fluorescently labeled RNA primer (Cy5-mRNA-prim, Supplementary Table S3) to enable RNA quantification and detection. For segmental modifying gene-coding part, while keeping the IRES region unmodified (Fig. 7A-3), we performed a two-step PEX. First, the IRES (Cy5-IRES-RNA,

Supplementary Table S3) was prepared by PEX with short RNA primer – Cy5-mRNA-prim and ssDNA template (templ_IRES, Supplementary Table S5, Supplementary Fig. S75) using natural rNTPs. Second, this RNA intermediate after template degradation was used as RNA megaprimer for second PEX using template – templ_IRES-nLuc and a mix of rATP, rUTP, rC$^{Me}$TP and rGTP. For preparation of point-modified mRNA (Fig. 7A-4, A-5) we applied a methodology combining SNI and PEX with two differently long ssDNA templates. First, using Cy5-IRES-RNA as RNA megaprimer and ssDNA template – templ_IRES-nLuc we performed SNI with rC$^{Me}$TP, followed by degradation of unincorporated nucleotide. Subsequent PEX with natural rNTPs generated full-length mRNA with modification in the initial part of the gene encoding sequence (Fig. 7A-4). Second, for preparation of modified

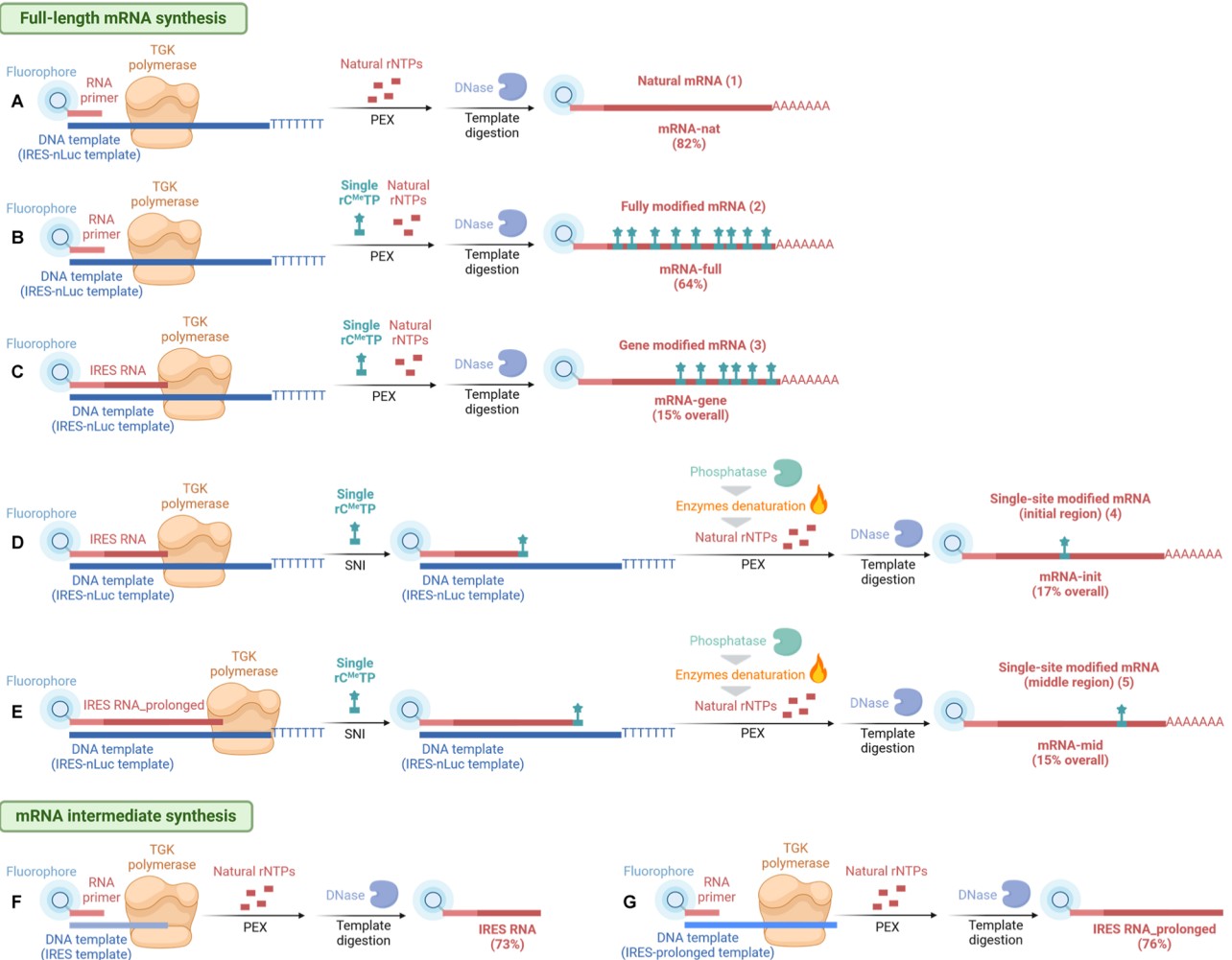

**Fig. 6 | Procedures for mRNA synthesis.** mRNA constructs were generally prepared by PEX with TGK polymerase and 5′-Cy5-labeled RNA primer and ssDNA template encoding for primer annealing site, IRES, gene-coding sequence and poly(A) tail. Several variations in the procedure, template or primer were implemented based on desired output. **A** Non-modified mRNA was prepared by PEX of RNA primer and template encoding for full-length mRNA with natural rNTPs. Natural mRNA (**A**) was isolated in 82% yield starting from 5′-Cy5-labeled RNA primer. **B** Fully modified mRNA was prepared by PEX, based on the same procedure, with combination of rATP, rUTP, rC^MeTP, rGTP. Fully modified mRNA (**B**) was isolated in 64% yield from 5′-Cy5-labeled RNA primer. **C** Gene-modified mRNA was prepared by PEX of longer IRES-spanning RNA megaprimer (**F**) and template encoding for full-length mRNA with combination of rATP, rUTP, rC^MeTP, rGTP. Gene-modified mRNA (**C**) was isolated in overall 15% yield after two steps, including intermediate synthesis (**F**), starting from 5′-Cy5-labeled RNA primer. **D, E** Single-site modified mRNAs

were prepared in the first step by SNI of rC^MeTP with RNA megaprimers (**F, G**) and template encoding for full-length mRNA followed by PEX in the second step with natural rNTPs. Point-modified mRNA with modification placed in the initial part of gene coding sequence (**D**) was prepared in overall 17% yield after two steps, including intermediate synthesis (**F**), starting from 5′-Cy5-labeled RNA primer. Point-modified mRNA with modification placed in the middle part of gene coding sequence (**E**) was prepared in overall 15% yield after two steps, including intermediate synthesis (**G**), starting from 5′-Cy5-labeled RNA primer. **F, G** RNA megaprimers were generated by PEX with 5′-Cy5-labeled RNA primer and template encoding for future unmodified mRNA part, i.e., IRES (**F**) or IRES and part of gene coding sequence (**G**). IRES-RNA was isolated in 73% yield from 5′-Cy5-labeled RNA primer (**F**). IRES-RNA_prolonged was isolated in 76% yield from 5′-Cy5-labeled RNA primer (**G**). Created with BioRender.com.

mRNA in the middle part of the gene encoding sequence (Fig. 7A-5) we applied similar methodology, only longer RNA megaprimer (Cy5-IRES-RNA_prolonged, Supplementary Table S3) was used in this case. All mRNAs and RNA megaprimer intermediates were verified by urea agarose gel (Supplementary Fig. S79) and single-point modified constructs also by sequencing of RT-PCR products from bisulfite-treated mRNA (Supplementary Figs. S263 and S264).

To evaluate translation efficiency of modified mRNA constructs, we subjected them to in vitro translation studies using rabbit reticulocyte lysate system. Intriguingly, both point-modified mRNAs (Fig. 7A-4, A-5) enhanced translation efficiency in comparison with natural mRNA (Fig. 7A-1), while the gene-modified mRNA (Fig. 7A-3) led to moderate decrease in potency. As expected, 5-methylcytosine substitutions in whole mRNA length (Fig. 7A-2) were detrimental to

translation (Fig. 7B, D), most likely due to IRES tertiary structure disruption and loss of ribosome binding. The best performing constructs – two mRNAs with single methyl-modification (Fig. 7A-4, A-5) were tested in *in cellulo* translation studies with HEK293T cells together with natural mRNA (Fig. 7A-1) as a control. Protein yields followed the same pattern as in in vitro experiments (Fig. 7C).

## Discussion

Herein we elaborated an efficient route to nucleobase-modified RNA that is based on previously reported[43–45] engineered thermophilic primer-dependent DNA polymerases. While by means of IVT with T7 RNAP it has been possible to install smaller, dispersed modifications along whole RNA length[30–32,34,35], we demonstrated superior performance of engineered thermophilic polymerases in several aspects

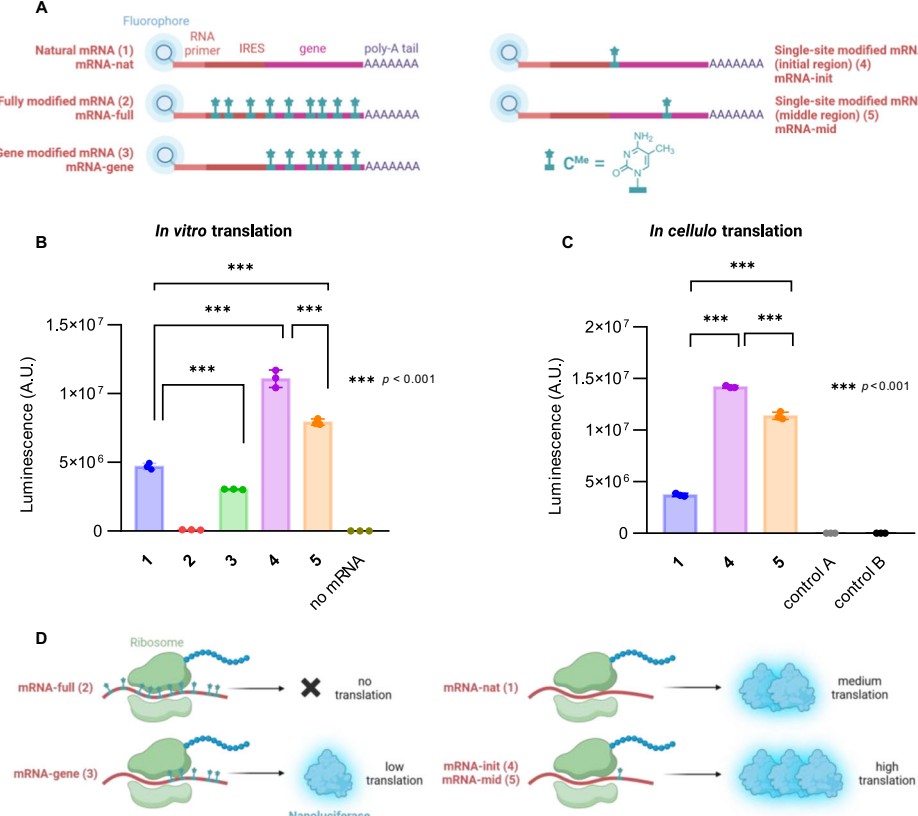

**Fig. 7 | Modified mRNA translation studies. A** Scheme of a set of natural or 5-methylcytosine modified mRNA molecules. Each molecule is composed of 5′-Cy5-labeled RNA primer, encephalomyocarditis virus internal ribosomal entry site (IRES), coding sequence for nanoluciferase and poly(A) tail. Different labeling strategies allowed to prepare natural mRNA (1), whole-length modified mRNA (2), gene modified mRNA (3), mRNAs bearing single modification in initial (4) or middle (5) part of the gene-coding sequence. For detailed synthesis of mRNA constructs see Fig. 6. **B** mRNA constructs and negative control without mRNA were subjected to in vitro translation with rabbit reticulocyte lysate system and probed for nanoluciferase activity. Statistical differences were evaluated with one-way ANOVA with Tukey–Kramer test for multiple comparison. Error bars represent standard deviations from mean of three independent biological replicates. *p*-values < 0.001. **C** In analogy to in vitro assay, constructs (1), (4) and (5) were transfected into

HEK293T cells, and upon 4 h incubation luciferase signal was measured. Control A represents signal from cells treated only with Lipofectamine MessengerMAX and NanoLuc substrate. Control B represents signal from non-treated cells (neither transfection agent nor mRNA) upon NanoLuc substrate addition. Statistical differences were evaluated with one-way ANOVA with Tukey–Kramer test for multiple comparison. Error bars represent standard deviations from mean of three independent biological replicates. *p*-values < 0.001. **D** Summary of luciferase translation assays outcome. Fully modified mRNA (2) yielded virtually no protein. Modifications in the gene-coding part of mRNA (3) allow for moderate translation efficiency, compared to natural mRNA (1) control. Single modifications introduced in either initial (4) or middle (5) region of the gene-coding part of mRNA elicit significant translation enhancement. Created with BioRender.com. Source data are provided as a Source Data File.

(Fig. 3, Supplementary Fig. S266). TGK polymerase, and to less extent SFM4-3 polymerase, incorporated bulky and hydrophobic groups at high density, as well as highly reactive CA or aldehyde (FT) moieties. Notably, we showed TGK polymerase, but not SFM4-3, can traverse precarious regions, such as U-rich homopolymers, where T7 RNAP terminates (Fig. 3). These findings are congruent to previous structural research on cognate thermophilic DNA polymerases. It revealed that B-family class, where TGK polymerase belongs to, is well adapted to accommodating major groove modifications attached to nucleobases[46]. On the other hand, SFM4-3 derived from A-family Taq polymerase, had substantially narrower scope, limited to smaller modifications in less dense arrangement. Facile incorporation of modified rG$^X$TPs, which is generally inefficient with T7 RNAP and requires unmodified initiator nucleotide[32], was achieved even for sterically demanding rG$^{Ph}$TP modification.

Previously elusive, high density all-four-nucleobase hypermodified RNA was obtained by TGK polymerase. To probe length limits we tested synthesis of fully modified RNA along different sized DNA templates to conclude, that at least 781-nt long RNA carrying modification on each nucleobase can be reliably obtained (Supplementary Figs. S45–S49, S80, S217–S224). We also demonstrated successful

reverse transcription to cDNA with SuperScript IV (Supplementary Fig. S54), which is important for directed evolution experiments with heavily modified RNA and allowed us to further substantiate correct length of modified RNA (Supplementary Figs. S81–S85).

Incorporation of modified substrates could putatively impair fidelity of the polymerase which might misincorporate other natural ribonucleotides in absence of the correct non-modified canonical nucleotide[55–57]. Therefore, we tested the polymerase fidelity by competitive incorporation analogously to previous studies[58]. Our results clearly showed that even a relatively poor and bulky substrate (e.g., rC$^{mBdp}$TP) is accepted with good fidelity despite presence of large excess of natural rNTPs (rATP, rUTP, rGTP), confirming that misincorporation with incorrect natural nucleotides is negligible. The synthesis fidelity of modified RNA was further confirmed by MS analysis of all oligonucleotides.

Generation of multiple copies of RNA from single dsDNA template molecule is significant advantage of T7 RNAP IVT. Here we showed, that also PEX can be cycled to analogously deliver multiple copies of RNA from one ssDNA template. The only pitfall to avoid was repeated denaturation and high-temperature incubation, harmful to long RNA, that we circumvented by immobilization of ssDNA template on solid

support, that enabled simultaneous RNA purification and recycling of the template and putatively also polymerase, nucleotides, and other reagents. It is worth to mention that cost-effective production of sufficient quantities of ssDNA is well established[59–61] and hence template recycling is not essential. Therefore, scaling up of PEX in large volume is a viable option, when inexpensive template can be used. Reasonable yields of modified RNA oligonucleotides can be obtained this way, further broadening usability of our method.

In analogy to principles of enzymatic DNA synthesis, our approach delivers flexible choice of primer chemistry and 5′-terminus labeling. This is a significant progress, since 5′-terminal RNA labeling is notoriously difficult issue. Current method for 5′-terminal labeling with T7 RNAP requires adding excess of expensive[62] or commercially unavailable[63] initiators. Additionally, the transcription efficiency then varies significantly, and labeling is never equimolar[64]. Unlike other prior approaches[65], our strategy does not require additional steps for post-synthetic labeling (e.g., RNA ligation or chemical reactions[66], that might be detrimental to sensitive modifications). Taking advantage of primer chemistry flexibility we devised a method, whereby synthetic DNA primer can be trimmed off with assistance from UDG with coincident DNA template digestion by TurboDNase to scarlessly leave behind RNA entirely built of modified nucleotides (Fig. 4A). Moreover, no expensive RNA oligonucleotide is then needed for priming. When slightly modified, this procedure allows to retain single fluorescently labeled primer-originating-dT nucleotide that facilitates tracking of the residual all-base modified RNA (Fig. 4D). Besides fluorophore, primer-originating-dT can bear variety of tags or labels, further expanding plausible functionalisation.

Decorating RNA at a designated internal position is largely unmet need that precludes effective study of RNA structure, interactions, dynamics and epitranscriptome[20]. To fill this methodological gap, we deployed improved version of stepwise SNI reaction previously developed for DNA labeling[49]. Here we demonstrated synthesis of dual- or triple-labeled riboswitch, which changed FRET in response to ligand addition (Fig. 5). Whole procedure allowed access to RNA labeled at user-defined positions, even inside homopolymeric sequence, with respectable yield (13%, Supplementary Fig. S73). IVT with T7 RNAP and modified rN^XTPs usually leads to pervasive RNA labeling. Also, introduction of multiple distinct or bulky modifications might be inefficient due to T7 RNAP limited tolerance. Single labeling using PLOR within homopolymeric sequences is not possible, and method requires programming of robotic platform for automation[67,68]. On contrary, our "one-pot" approach occurring in a single tube is virtually nondependent on desired modified position and allowed us to instal simultaneously two bulky Cy5 and Cy3 fluorophores. Since additional tag was brought in as a part of synthetic primer, up to three labels were collectively introduced in the riboswitch molecule.

Single nucleotide labeled mRNA would be attractive tool to investigate position-dependent effects of nucleobase modifications, however, most available approaches allow to examine only uniformly modified RNA. Site-specific labeling with PLOR suffers from decreased yields, when the label is distant from 5′-end due to increasing number of pause-restart cycles and washing steps, preventing synthesis of very long RNAs. Only one procedure has been described to allow installation of single-nucleotide modification within long RNA. Exploiting ligation of synthetic and enzymatically generated fragments provided access to mRNA with one 2′-O-methyl nucleotide, but this method suffered from poor yields (3–7%) and tedious gel purifications[25], or restricted labeling position to be in close distance from mRNA termini[69]. Our strategy allows to decorate mRNA with region-targeted or single-site modifications (Figs. 6 and 7) and full-length products are isolated by means of poly(T)-beads capture, that can be readily automated to allow high-throughput screening. Unlike conventional T7 RNAP IVT[52], our method obviates need to HPLC purify mRNA, since no dsRNA by-products can be generated. Here, we for the first time

revealed that single 5-methylcytosine modifications within gene-coding region deliver enhanced protein expression not only in reticulocyte lysate, but also upon transfection in HEK293T cells (Fig. 7B–D). This clearly underscores importance of determining not only type but also sequence context of modifications within mRNA and opens door to discovery of improved mRNA therapies.

To conclude, here we disclose a general and efficient strategy for synthesis of on-demand modified RNA based on employing engineered thermostable DNA polymerases. We demonstrate superior performance of PEX with these enzymes over IVT with T7 RNAP in accepting wide range of natural epitranscriptomic, reactive, hydrophobic, fluorescent, or affinity-tagged rN^XTPs, including multiple combinations thereof, leading to even highly functionalized or hypermodified RNA. Also, integration of synthetic oligonucleotide RNA primer bearing additional functionalities is possible, but not essential. On the other hand, DNA-primed synthesis enables traceless primer removal, 5′-terminal labeling, or preparation of unique DNA-RNA hybrids. Furthermore, we conducted multisite-specific labeling of highly structured molecules with distinct tags for functional studies. Our method overcomes several shortcomings of previous approaches and makes progress towards synthesis of long modified RNA. Leveraging our method, we unraveled importance of single-modification positioning within mRNA that led to significant enhancement of translation. Multiple applications in RNA biochemistry or biotechnology, as well as in production of potential modified RNA therapeutics are envisaged and will be pursued in our laboratory.

## Methods

### Preparation of all-nucleobase modified RNA *via* primer removal
Reaction was performed in total volume of 10 μL in ThermoPol buffer (1X). ssDNA template – templ_35nt (4.8 μM) and dually labeled DNA primer with 5′-(Cy5) and internal dU and dT-(FAM) modification – Cy5-DNA-prim_dU_dT-FAM (4.0 μM) in ThermoPol buffer were heated up to 95 °C for 30 s and then cooled down to 3 °C (0.1 °C s⁻¹). After addition of TGK polymerase (2.0 μM) and a mixture of rA^ETP, rU^BioTP, rC^PhTP, rG^PentTP (0.2 mM), the reaction was incubated at 60 °C for 2 h in a thermal cycler with heated lid (100 °C). Positive control was performed under same conditions with a mixture of natural rNTPs (0.2 mM) instead of the modified rN^XTPs. Five different negative control reactions were performed under same conditions using in each reaction a different mixture of three natural rNTPs [0.2 mM; r(AUC), r(AUG), r(ACG), r(UCG)] and H₂O or by complete replacement of natural rNTPs by H₂O. For sample preparation for dPAGE analysis see protocol in Supplementary Information Section 2.5.1. For cleavage of the DNA primer, the crude mixture (35DNA_dU_dT^FAM-RNA_A^EU^BioC^PhG^Pent) was combined with UDG buffer (1 μL, 10X), UDG (2 U) and further incubated at 37 °C for 30 min. Solution of DMEDA (1.2 μL, 1 M, pH 8.5) was added and again incubated at 37 °C for further 30 min to generate the desired cleavage product (dT^FAM-RNA_A^EU^BioC^PhG^Pent). After that, TurboDNase (2 U) was added and incubated at 37 °C for 1 h. For sample preparation for dPAGE analysis see protocol in Supplementary Information Section 2.5.1. For dPAGE analysis of PEX see Supplementary Fig. S70 and for UDG cleavage see Supplementary Fig. S71.

### Preparation of IRES fragment
Reaction was performed in total volume of 10 μL in ThermoPol buffer (1X). The ssDNA template – templ_IRES (0.24 μM, prepared according to Supplementary Information Section 2.19.1) and 5′-(Cy5)-labeled RNA primer – Cy5-mRNA-prim (0.2 μM) in presence of ThermoPol buffer were heated up to 95 °C for 30 sec and then cooled down to 3 °C (0.1 °C s⁻¹). After addition of a mixture of rNTPs (0.4 mM) and TGK polymerase (0.1 μM) the reaction was incubated at 65 °C for 4 h, followed by final work up using the standard protocol in Supplementary Information Section 2.5.19. For generation of larger quantities of IRES-RNA, the reaction was 10X scaled-up. For sample preparation protocol

for gel analysis see Supplementary Information Section 2.5.2. For urea agarose gel analysis see Supplementary Fig. S79.

## Synthesis of single-site modified mRNA (mRNA-init)

Reaction mixture (10 μL) containing ThermoPol buffer (1X), rC^MeTP (0.2 mM), ssDNA template – templ_IRES-nLuc (0.24 μM, prepared according to Supplementary Information Section 2.19.3), 5´-(Cy5)-labeled RNA primer – Cy5-IRES-RNA (0.2 μM, IRES fragment, prepared according to above mentioned procedure) and TGK polymerase (0.75 μM) was heated up to 95 °C for 30 s followed by incubation at 65 °C for 20 min. After this, the mixture was combined with rSAP (1 U) and incubated at 37 °C for 30 min, followed by enzyme denaturation at 65 °C for 5 min. Then a mixture of rNTPs (4 mM, 1 μL) was added and the reaction was further incubated at 65 °C for 4 h, followed by final work up using the standard protocol in Supplementary Information Section 2.5.20. For generation of larger quantities of mRNA-init the reaction was 3X scaled-up. For sample preparation protocol for gel analysis see Supplementary Information Section 2.5.2. For urea agarose gel analysis see Supplementary Fig. S79.

## In vitro translation in rabbit reticulocyte lysate system

Reaction mixture (10 μL) containing 50 ng of mRNA, Ribolock RNase inhibitor (40 U), complete amino acids mixture (50 μM) and nuclease-treated rabbit reticulocyte lysate was incubated at 30 °C for 1.5 h in a thermal cycler with heated lid (65 °C). Translation reactions were stopped by addition of cycloheximide solution (100 μM, in DMSO). 1 μL aliquot of the quenched reaction mixture was combined with 20 μL of reconstituted Nano-Glo luciferase assay reagent prepared by combining Nano-Glo luciferase assay substrate and Nano-Glo luciferase assay buffer (1:50) according to manufacturer´s protocol. The mixture was incubated at room temperature for 3 min and 20 μL aliquots were transferred to white opaque 384-well microplate prior to luminescence measurements with 1000 ms integration time on Tecan Spark Multimode Reader. The data are reported as counts/sec. For negative control, mRNA was omitted.

Details for other methods are included in Supplementary Information file.

## Reporting summary

Further information on research design is available in the Nature Portfolio Reporting Summary linked to this article.

## Data availability

The data supporting the findings of this study are available from the corresponding authors upon request. Source data for the figures and Supplementary Figs. are provided as a Source Data file. Source data are provided with this paper.

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

## Acknowledgements

Funding by Czech Science Foundation (20-00885X to M. B., M. K. and M. H.) is gratefully acknowledged. Additional experiments performed in the revision stage were funded by the Ministry of Education, Youth and Sports of the Czech Republic grant RNA for therapy (CZ.02.01.01/00/ 22_008/0004575 to M. B. and M. H.). Authors thank K. Nováková and E. Kofroňová for MS-MALDI measurements, K. Kertisová for small molecule HR-ESI-MS measurements, H. Cahová for critical reading of the manuscript, J. C. Chaput for kind provision of pGDR11-TGLK plasmid and F. E. Romesberg for pET-SFM4-3 plasmid. All figures except of Fig. 2 were created with BioRender.

## Author contributions

M.H. set the overall direction of research and secured funding; M.B., M.K. and M.H. conceived and designed experiments; M.B. performed all

experimental work apart from that carried out by other researchers as indicated in following: V.H. synthesized hydrophobic and clickable nucleotides; J.M. synthesized rC$^{mBdp}$TP; V.H. and M.K. expressed and purified SFM4-3 polymerase; R.P. and L.P.S. performed NMR spectra measurements and assignments; M.K. performed *in cellulo* translation studies and RT-qPCR analysis; M.B., M.K. and M.H. analyzed and interpreted data and wrote the manuscript with inputs from all other authors.

## Competing interests

The authors declare no competing interests.
