## [Peer Review File · Nature Communications]

Expedient production of site specifically nucleobase-labelled or hypermodified RNA with engineered thermophilic DNA polymerasesREVIEWER COMMENTS

Reviewer #1 (Remarks to the Author):

In this article, the authors report a novel method to produce RNAs equipped with chemical modifications at user-defined positions based on thermostable, engineered DNA polymerases. The authors first investigated the possibility of introducing modified RNA nucleotides into RNA with two engineered DNA polymerases and compared the efficiency of the process to that of standard RNA polymerases. Since enzymatic synthesis relies on primer extension reactions, the authors then developed a highly efficient method for DNA primer removal. The combination of DNA primer removal and PEX reactions with the engineered DNA polymerases allowed even for the incorporation of four modified RNA nucleotides. Lastly, the authors devised a new strategy for the implementation of modifications at virtually any position of RNA sequences and used this strategy to produce mRNAs equipped with one or multiple modifications and evaluated the effect of the modification (in this case 5-Me-cytosine) on translation. All experiments were carried out in a competent manner, the manuscript is well-written and clear, and the supporting information is of high quality. Overall, the method described herein will be very useful for identifying novel modifications in RNA therapeutics in general and publication in Nature Communications is highly recommended pending some (minor) revisions:

- The authors clearly demonstrate that this method is highly efficient and versatile for base-modified nucleotides, but can it be extended to the production of RNAs containing a mixture of base, sugar, and phosphate modifications since these engineered polymerases were evolved to tolerate modifications at 2'?
- Line 117: the authors mention that for CA-modified nucleotides side-conjugation reactions were observed. The authors should probably give a short description of the type of side-reactions that can occur with these modifications.
- The authors may want to introduce a more detailed scheme in the ESI summarizing the synthetic methods described in Figure 6A.

Reviewer #2 (Remarks to the Author):

Within the submitted manuscript Hocek and colleagues present an innovative and highly promising strategy for the controlled synthesis of nucleobase-modified RNA, addressing a critical need in the field of RNA biology exploration and therapeutic development. The authors have utilized primer-dependent engineered thermophilic DNA polymerases, specifically SFM4-3 (originally developed by the Romesberg group) and TGK (originally developed by the Holliger group), to achieve the introduction of various base-modified nucleotides in a single RNA strand. This approach not only allows for the synthesis of hypermodified RNA containing all four modified nucleotides with distinct substituents but also outlines a method for primer segment removal.

The manuscript showcases the versatility of the proposed strategy by demonstrating the facile site-specific or segmented introduction of fluorophores and other functional groups at defined positions within a range of RNA molecules. This includes structured or long mRNA, indicating the broad applicability and flexibility of the presented approach.

One of the notable strengths of this work is the emphasis on the translation efficacy of single-site modified mRNAs. The observation of intriguing translation efficacy underscores the importance of studying isolated modifications placed at designer positions. This is a crucial step in unraveling their biological effects, providing essential insights for the development of improved mRNA therapeutics. The toolbox introduced in this manuscript not only expands the capabilities for dissecting RNA structures and functions but also holds significant promise for constructing diverse types of base-functionalized RNA with potential applications in therapeutic interventions and diagnostics. The manuscript successfully communicates the potential impact of this novel approach on advancing our understanding of RNA biology and facilitating the development of RNA-based therapeutics.

Overall, the quality of the research, the clarity of presentation, and the potential impact on the field could make this manuscript highly suitable for publication in Nature Communications. However, on the basis of the data presented, I believe that the research is too premature to warrant publication in its current form, as two very important aspects still need to be addressed:

1) An important advantage of the method shown here is that highly modified RNA (i.e. carrying a modification at each base) can be synthesised. However, this is only shown for the synthesis of rather short oligonucleotides. mRNA vaccines, a field of application mentioned by the authors, require long RNA strands. Therefore, in my opinion, it is necessary to investigate which lengths of fully modified RNA are accessible. This could be investigated, for example, by performing a meaningful PEX analogous to the experiments shown in Figure 3-5 using M13mp18 single-stranded DNA as template.

2) One advantage of IVT is that the DNA containing the promoter and template only needs to be present in sub-stoichiometric concentrations compared to the mRNA obtained (i.e. several mRNA molecules can be obtained from one template molecule). This turnover aspect is particularly important for obtaining large amounts of mRNA in a cost-effective manner. This aspect is not addressed in the manuscript. Is it also possible to obtain several RNA molecules from one template molecule with the presented approach?

Reviewer #3 (Remarks to the Author):

Synthesis of custom functionalized long RNA sequences is highly important to advance research in RNA biology and therapeutics. However, it remains a major challenge. In this context, Hocek and coworkers have elegantly repurposed two engineered thermophilic DNA polymerases, namely TGK and SFM4-3 (reported earlier) to enzymatic construct nucleobase-modified RNAs based on primer-extension (PEX) reaction. The authors have designed different PEX approaches to install a variety of nucleobase modified nucleotide analogs at different locations within the RNA strand. In particular, reactions with TGK produced hypermodified RNA containing all four modified nucleotides. Site-specific or segmented introduction of fluorophores is also described, which was further put to use in synthesizing FRET-pair labeled adenine riboswitch. Lastly, authors introduced 5-methylcytosine site-specifically using their protocol and showed that thus obtained mRNA exhibited enhanced protein expression in lysate and in cells.

Engineered thermophilic DNA polymerases and strategies to functionalize RNA described in this manuscript are very useful and important. Experiments have been nicely conceived and systematically, and adequate characterization methods have been used to study the products. This manuscript is suitable for publication in Nature Communications. I have few important comments and suggestions, which can be incorporated in the manuscript.

Specific comments:

1. Line 110 and reactions related to PEX: It would be convenient to include in the text, which primer is included in the reaction. The authors mention the template used, but not always the primer.

2. Supplementary Figures S6-S30: In the absence of respective natural nucleotide or modified nucleotide, there is some amount (sometimes significant amounts) of full length product formation due to misincorporation. For eg., see Fig S6, S12, S13, S14, S16 etc. This means the fidelity of the enzymes is compromised when using certain nucleotides (e.g., C). This could also happen in actual reactions wherein modifications are introduced, which would affect the sequence integrity. The authors should elaborate on these observations.

3. Line 149: Table numbering is not right.

4. One important problem with PEX as compared to IVT, is the amount of product that can be obtained

by PEX is lower. In all incorporation reactions, author analyze by gel electrophoresis, but isolated yields are not provided. Can large-scale reactions can be done using these engineered polymerases, if so the scale and yields. The authors can show using a few of the nucleotides analogs. Such data will enhance the practical scope of the enzyme and reactions described in this manuscript.

5. The supporting information is very huge. I see that the reactions conditions can be provided in the form of a Table after giving a general procedure for each type of a reaction.

Response to reviewers and list of changes

REVIEWER COMMENTS

Reviewer #1 (Remarks to the Author):

In this article, the authors report a novel method to produce RNAs equipped with chemical modifications at user-defined positions based on thermostable, engineered DNA polymerases. The authors first investigated the possibility of introducing modified RNA nucleotides into RNA with two engineered DNA polymerases and compared the efficiency of the process to that of standard RNA polymerases. Since enzymatic synthesis relies on primer extension reactions, the authors then developed a highly efficient method for DNA primer removal. The combination of DNA primer removal and PEX reactions with the engineered DNA polymerases allowed even for the incorporation of four modified RNA nucleotides. Lastly, the authors devised a new strategy for the implementation of modifications at virtually any position of RNA sequences and used this strategy to produce mRNAs equipped with one or multiple modifications and evaluated the effect of the modification (in this case 5-Me-cytosine) on translation. All experiments were carried out in a competent manner, the manuscript is well-written and clear, and the supporting information is of high quality. Overall, the method described herein will be very useful for identifying novel modifications in RNA therapeutics in general and publication in Nature Communications is highly recommended pending some (minor) revisions:

Our response: We thank the reviewer for the very positive assessment of our work.

Question: 1) The authors clearly demonstrate that this method is highly efficient and versatile for base-modified nucleotides, but can it be extended to the production of RNAs containing a mixture of base, sugar, and phosphate modifications since these engineered polymerases were evolved to tolerate modifications at 2'?

Our response: We thank the reviewer for this valuable and logical advice. We are aware of the possibilities provided by these engineered enzymes for synthesis of ribose-modified XNAs, since actually both polymerases have been developed and studied for incorporation of 2'-modified nucleotides. On the other hand, the suggested enzymatic synthesis of phosphate-modified nucleic acids would probably be very problematic because previous work by the Holliger lab showed that polymerases accepting larger phosphate modifications (e.g., methylphosphonates) are of poor efficiency (*Nat. Chem.* **2019**, *11*, 533–542), and would probably require further mutagenesis and engineering to perform well with combined phosphate-nucleobase modifications.

We consider these suggested advanced applications to be far beyond the scope of this manuscript focused solely on base-modified RNA, which we believe is novel and interesting on itself. The paper in the current scope is already very large and information-rich and the enzymatic synthesis of dual base- and sugar-modified RNA will be an object of a separate study and separate publication in the future.

Changes made: none

Question: 2) Line 117: the authors mention that for CA-modified nucleotides side-conjugation reactions were observed. The authors should probably give a short description of the type of side-reactions that can occur with these modifications.

Our response: We would kindly bring to reviewer's attention that the structures of the by-products have been already depicted in supplementary data file together with the mass spectra of these

species (Supplementary Figures S116, S117, S134, S135, S145, S146, S161, S162 in the original version of Supplementary Data). Unfortunately, due to complex macromolecular nature of the RNA oligonucleotides, we are unable to determine exact site of side reactions on the target nucleobase (e.g., using NMR). On the other hand, we would like to emphasize that we were able to optimise conditions to suppress these unwanted intramolecular cyclisation side reactions, hence this issue is not limiting the applicability of our method at all.

Changes made: In order to further clarify this, we added an additional scheme depicting the nature of the side-products to Supporting Information as Figure S129, S130, S147, S148, S158, S159, S174, S175 in the revised Supplementary Data file.

Question: 3) The authors may want to introduce a more detailed scheme in the ESI summarizing the synthetic methods described in Figure 6A.

Our response: We would kindly bring to reviewer's attention that detailed description of reactions and synthetic schemes for the production of modified mRNA have been already enclosed in our original manuscript as Extended Data Figure 2. In the caption of Figure 6A, we have statement: "For detailed synthesis of mRNA constructs see Extended Data Fig. 2".

Changes made: none

Reviewer #2 (Remarks to the Author):

Within the submitted manuscript Hocek and colleagues present an innovative and highly promising strategy for the controlled synthesis of nucleobase-modified RNA, addressing a critical need in the field of RNA biology exploration and therapeutic development. The authors have utilized primer-dependent engineered thermophilic DNA polymerases, specifically SFM4-3 (originally developed by the Romesberg group) and Tgk (originally developed by the Holliger group), to achieve the introduction of various base-modified nucleotides in a single RNA strand. This approach not only allows for the synthesis of hypermodified RNA containing all four modified nucleotides with distinct substituents but also outlines a method for primer segment removal.

The manuscript showcases the versatility of the proposed strategy by demonstrating the facile site-specific or segmented introduction of fluorophores and other functional groups at defined positions within a range of RNA molecules. This includes structured or long mRNA, indicating the broad applicability and flexibility of the presented approach.

One of the notable strengths of this work is the emphasis on the translation efficacy of single-site modified mRNAs. The observation of intriguing translation efficacy underscores the importance of studying isolated modifications placed at designer positions. This is a crucial step in unraveling their biological effects, providing essential insights for the development of improved mRNA therapeutics.

The toolbox introduced in this manuscript not only expands the capabilities for dissecting RNA structures and functions but also holds significant promise for constructing diverse types of base-functionalized RNA with potential applications in therapeutic interventions and diagnostics. The manuscript successfully communicates the potential impact of this novel approach on advancing our understanding of RNA biology and facilitating the development of RNA-based therapeutics.

Overall, the quality of the research, the clarity of presentation, and the potential impact on the field could make this manuscript highly suitable for publication in Nature Communications.

Our response: We thank the reviewer for the very positive assessment of our work.

However, on the basis of the data presented, I believe that the research is too premature to warrant publication in its current form, as two very important aspects still need to be addressed:

Question: 1) An important advantage of the method shown here is that highly modified RNA (i.e. carrying a modification at each base) can be synthesised. However, this is only shown for the synthesis of rather short oligonucleotides. mRNA vaccines, a field of application mentioned by the authors, require long RNA strands. Therefore, in my opinion, it is necessary to investigate which lengths of fully modified RNA are accessible. This could be investigated, for example, by performing a meaningful PEX analogous to the experiments shown in Figure 3-5 using M13mp18 single-stranded DNA as template.

Our response: We thank for these valuable suggestions. In our manuscript we explored inclusion of all-nucleobase modifications only with shorter RNA, since we envisage putative applications predominantly in aptamers or shorter functional RNAs. For long RNAs, typically one or few modifications will be used – as we and others (*ChemBioChem* **2023**, *24*, e202200658) have shown that grafting high density of modifications along whole mRNA, even on single nucleobase type (i.e., 5-methylcytosine), is very likely detrimental to mRNA translation. Therefore, we developed and implemented a method for single-site labelling, which seems to be promising for further lines of our research. However, we admit, that in certain settings, obtaining long RNAs with multiple modifications could be inspirational for other researchers. For that purpose, we carried out additional experiments to prove length limits of heavily modified RNA synthesis. We synthesised and visualised RNAs carrying modifications on every nucleobase up to 804 nucleotides long (781 of them carrying modification) in agarose gels. That shall be already sufficient length encoding 268 amino acid long protein (if IRES is unmodified or replaced by cap). For 1147nt-long strand, corresponding to our IRES-nLuc mRNA, we have already seen smearing in the agarose gel, presumably due to progressive alteration of mRNA biophysical properties affecting migration in the gel. As a workaround, we performed RT-qPCR to detect full-length products for all RNAs we synthesized. Gel analysis confirmed presence of full-length products of expected length, while control reactions without reverse transcription produced virtually no product.

Changes made: In main text we added the following paragraph:

*To probe length limitations of TGK-mediated synthesis of hypermodified RNA, we conducted PEX with either natural rNTPs or a mix of rA^ETP, rU^ETP, rC^{Me}TP, rG^{Pent}TP on ssDNA templates – **templ_IRES**, **templ_IRES-prolonged**, **templ_IRES-nLuc** (Supplementary Table S5, Supplementary Fig. S75-S77) varying in length (encoding for 606, 804 or 1147-nt long RNAs) and **Cy5-mRNA-prim**. Agarose electrophoresis revealed that **templ_IRES** and **templ_IRES-prolonged** produced full-length products with both natural and modified nucleotides (Supplementary Fig. S80). The longest template – **templ_IRES-nLuc** gave sharp band of full-length product in case of natural rNTPs, but only smeared band was observed for modified rN^XTPs (Supplementary Fig. S80). To confirm formation of products of correct length, we carried out reverse transcription and subsequent qPCR analysis to ensure that only cDNA (and not DNA template) is amplified (Supplementary Fig. S82, S84, S85). Bands of expected lengths were obtained for all three modified RNA templates (Supplementary Fig. S81 and S83).*

and to the Discussion section:

To probe length limits we tested synthesis of fully modified RNA along different sized DNA templates to conclude, that at least 781-nt long RNA carrying modification on each nucleobase can be reliably obtained (Supplementary Fig. S45-S49, S80, S217-S224).

...and allowed us to further substantiate correct length of modified RNA (Supplementary Fig. S81-S85).

Additional sections describing experimental conditions for PEX reactions with long templates were added as sections 2.19.15 (PEX reactions), 2.19.16 and 2.19.17 (reverse transcriptions), 2.19.18 and 2.19.19 (qPCR and gel analysis) to Supplementary Data file. Additional figures of gel analysis of PEX products (Supplementary Fig. S80), qPCR curves (Supplementary Fig. S82, S84, S85) and gel analysis of re-amplified DNA products (Supplementary Fig. S81, S83) were added to Supplementary Data file.

Question: 2) One advantage of IVT is that the DNA containing the promoter and template only needs to be present in sub-stoichiometric concentrations compared to the mRNA obtained (i.e. several mRNA molecules can be obtained from one template molecule). This turnover aspect is particularly important for obtaining large amounts of mRNA in a cost-effective manner. This aspect is not addressed in the manuscript. Is it also possible to obtain several RNA molecules from one template molecule with the presented approach?

Our response: We acknowledge that state-of-the-art methods for mRNA manufacturing are based on *in vitro* transcription of linearised plasmids with T7 RNAP. However, we would like to stress there are state-of-the-art methods to generate and obtain large amounts of ssDNA at a reasonable cost by PCR followed by strand separation, asymmetric PCR or M13 bacteriophage cultivation (e.g., *Nucleic Acids Res.* **2019**, *47*, 11956–11962; *Sci. Rep.* **2019**, *9*, 1–9; *Nature* **2017**, *552*, 84–87). Hence template production is not in our opinion significantly limiting factor of the workflow.

Despite of this, we consider template recycling to be an interesting question to answer. To figure it out, we carried out repeated PEX reaction, analogously to asymmetric PCR, to unpick, whether we can generate more nucleobase-modified RNA molecules from single template ssDNA strand. To our satisfaction, we observed much more product in cycled (denatured-annealed-extended) reactions, compared to single PEX extensions. Unfortunately, we also detected progressive RNA fragmentation. This suggests that this method, unlike for DNA, will have certain limitations for sensitive modifications and for longer RNA strands, that might be prone to thermal decay. To resolve this issue, we explored repeated PEX reactions with template immobilized via 5'-terminal dual-biotin onto streptavidin magnetic particles. After each extension, the product was liberated by mild NaOH denaturation and beads with template were reused for next PEX reactions. Although yields were gradually decreasing, we confirmed our ability to generate super-stoichiometric amount of RNA from single DNA template. We suggest, that for large-scaled production, acrylamide-anchored template (*Nucleic Acids Res.* **2019**, *47*, 11956–11962) could be analogously used and recycled to circumvent the need for streptavidin magnetic particles. This will be a subject of our further studies.

Changes made: We mentioned possibility of amplification PEX and template recycling in the main text:

Since T7 RNAP can generate multiple copies of RNA from single DNA molecule, we were also interested in possibility of template recycling for PEX. We repeatedly denatured, annealed, and extended primer **Cy5-RNA-prim_15nt** with ssDNA template **templ_98nt** (Supplementary Table S5) and either natural rNTPs or a mixture of rATP, rUTP, rC^{Me}TP, rGTP. As a control, non-cycled reaction with

single extension was carried out. Gel analysis revealed that cycling generates significantly higher amount of full-length product, however, also competing degradation of RNA was detected (Supplementary Fig. S88). Different strategy was needed for longer RNA due to their higher propensity to decay. The 1163-nt long dual-biotinylated ssDNA template – **templ_(dual-Bio)-IRES-nLuc** (Supplementary Table S5, Supplementary Fig. S78) immobilised onto streptavidin magnetic beads was successfully reused up to 5 times in solid-phase PEX with 23-nt long fluorescent RNA primer – **Cy5-mRNA-prim** (Supplementary Table S3) and either natural rNTPs or a mixture of rATP, rUTP, **rC^{Me}TP**, rGTP. The RNA products **mRNA-nat** or **mRNA-full** were liberated with mild NaOH elution after each PEX cycle, albeit with gradually decreasing yields as determined by agarose gel electrophoresis (Supplementary Fig. S86 and S87).

and to the Discussion section:

Generation of multiple copies of RNA from single dsDNA template molecule is significant advantage of T7 RNAP IVT. Here we showed, that also PEX can be cycled to analogously deliver multiple copies of RNA from one ssDNA template. The only pitfall to avoid was repeated denaturation and high-temperature incubation, harmful to long RNA, that we circumvented by immobilisation of ssDNA template on solid support, that enabled simultaneous RNA purification and recycling of the template and putatively also polymerase, nucleotides, and other reagents. It is worth to mention that cost-effective production of sufficient quantities of ssDNA is well established^{59,60,61} and hence template recycling is not essential.

Sections describing experimental details for repeated PEX reactions on shorter 98-nt long template have been added to Supplementary Data file as section 2.21 together with corresponding Supplementary Fig. S88. Repeated PEX reactions showing template recycling for longer RNA have been added to Supplementary Data file as sections 2.19.20 and 2.19.21 together with corresponding Supplementary Fig. S86 and S87.

Reviewer #3 (Remarks to the Author):

Synthesis of custom functionalized long RNA sequences is highly important to advance research in RNA biology and therapeutics. However, it remains a major challenge. In this context, Hocek and coworkers have elegantly repurposed two engineered thermophilic DNA polymerases, namely TGK and SFM4-3 (reported earlier) to enzymatic construct nucleobase-modified RNAs based on primer-extension (PEX) reaction. The authors have designed different PEX approaches to install a variety of nucleobase modified nucleotide analogs at different locations within the RNA strand. In particular, reactions with TGK produced hypermodified RNA containing all four modified nucleotides. Site-specific or segmented introduction of fluorophores is also described, which was further put to use in synthesizing FRET-pair labeled adenine riboswitch. Lastly, authors introduced 5-methylcytosine site-specifically using their protocol and showed that thus obtained mRNA exhibited enhanced protein expression in lysate and in cells.

Engineered thermophilic DNA polymerases and strategies to functionalize RNA described in this manuscript are very useful and important. Experiments have been nicely conceived and systematically, and adequate characterization methods have been used to study the products. This

manuscript is suitable for publication in Nature Communications. I have few important comments and suggestions, which can be incorporated in the manuscript.

Our response: We thank the reviewer for the very positive assessment of our work.

Specific comments:

Question: 1) Line 110 and reactions related to PEX: It would be convenient to include in the text, which primer is included in the reaction. The authors mention the template used, but not always the primer.

Our response: We thank for an apt recommendation.

Changes made: We have added this information accordingly.

Question: 2) Supplementary Figures S6-S30: In the absence of respective natural nucleotide or modified nucleotide, there is some amount (sometimes significant amounts) of full length product formation due to misincorporation. For eg., see Fig S6, S12, S13, S14, S16 etc. This means the fidelity of the enzymes is compromised when using certain nucleotides (e.g., C). This could also happen in actual reactions wherein modifications are introduced, which would affect the sequence integrity. The authors should elaborate on these observations.

Our response: We appreciate a fitting remark, that appears to require further clarification. In our experience negative controls are often producing false positive results. This is because extremely imbalanced mixture of nucleotides, where one of them is missing, renders polymerases to be excessively prone to errors (*J. Biol. Chem.* **1992**, 267, 3589–3596). This phenomenon is sometimes even deliberately used to enable error determination (*Nucleic Acids Res.* **2018**, 46, e78–e78.) or for *in vitro* mutagenesis (*Gene* **1990**, 88, 107–111.). For all of our synthesized modified RNA oligonucleotides we have carried out MS-confirmation of correct mass and hence also sequence. To further illuminate, whether nucleobase modifications can impair polymerase fidelity, we performed competitive incorporation PEX experiment, where bulky single modified nucleotide $rC^{mBdp}TP$ was to be incorporated in presence of excess of other three natural nucleotides (rATP, rUTP, rGTP). Different shifts due to modification size enabled to resolve correct products (containing rC^{mBdp}) from the faulty ones (containing rA, rU or rG). We have seen that even in 6x excess of “wrong” nucleotides, only traces of faulty products are generated.

In our real reactions we use generally balanced nucleotide compositions, therefore natural rNTPs misincorporations, even with such a poor substrate as $rC^{mBdp}TP$, are negligible and do not prevent any reasonable applications.

Changes made: We added a paragraph on the polymerase fidelity issue in the main text:

*In certain cases, the negative control experiments performed in absence of one natural rNTP showed some formation of spurious full-length product (e.g. Supplementary Fig. S12-S16) presumably due to misincorporation. For this reason, we have further tested the polymerase fidelity for incorporation of modified rN^xTPs . We performed single nucleotide extension of **Cy5-RNA-prim_15nt** complementary to **templ_16nt** encoding for one rC nucleotide with exceptionally bulky $rC^{mBdp}TP$ mixed at different ratios with three natural nucleotides (rATP, rUTP, rGTP) to determine misincorporation ratios. The use of this bulky nucleotide enabled clear distinction of the modified (correct product) and non-modified RNA (faulty product) on dPAGE. Gratifyingly, even with 6 equiv. excess of natural nucleotides (rATP, rUTP, rGTP) towards $rC^{mBdp}TP$, more than 90% of the product contained correct*

rC^{mBdp} nucleotide as determined by gel analysis, confirming sufficient fidelity of the synthesis of base-modified RNA (Supplementary Fig. S64 and S65).

and in Discussion section

Incorporation of modified substrates could putatively impair fidelity of the polymerase which might misincorporate other natural ribonucleotides in absence of the correct non-modified canonical nucleotide^{55,56,57}. Therefore, we tested the polymerase fidelity by competitive incorporation analogously to previous studies⁵⁸. Our results clearly showed that even a relatively poor and bulky substrate (e.g., rC^{mBdp}TP) is accepted with good fidelity despite presence of large excess of natural rNTPs (rATP, rUTP, rGTP), confirming that misincorporation with incorrect natural nucleotides is negligible. The synthesis fidelity of modified RNA was further confirmed by MS analysis of all oligonucleotides.

A section describing experimental details for determination of polymerase error rate via competitive incorporation has been added to Supplementary Data file as section 2.16.8 together with corresponding Supplementary Fig. S64, S65.

Question: 3) Line 149: Table numbering is not right.

Our response: After thorough inspection of numbering, we are not aware of any error in this particular section of the originally submitted manuscript.

Changes made: none

Question: 4) One important problem with PEX as compared to IVT, is the amount of product that can be obtained by PEX is lower. In all incorporation reactions, author analyze by gel electrophoresis, but isolated yields are not provided. Can large-scale reactions can be done using these engineered polymerases, if so the scale and yields. The authors can show using a few of the nucleotides analogs. Such data will enhance the practical scope of the enzyme and reactions described in this manuscript.

Our response: We agree that scalability might be an important factor affecting scope of future applications of our method. In general, volume of PEX reactions can be increased, as long as required amounts of reagents are supplied and attainable. Engineered polymerases can be expressed in large quantities from bacterial systems in cost-effective manner and nucleotides are available through chemical synthesis in reasonable amounts, even though some tweaks to the protocols could be necessary in some cases.

To demonstrate scale-up, we performed PEX reactions in 1 nanomolar scale, purified mixtures with silica-based columns, confirmed by MS analysis and quantified products by gel densitometry against standard dilution curve. We were able to obtain modified RNA oligonucleotides in good yields ranging from 63% to 78%, depending on modification type. We envisage, that further scale up can be done, but for most biochemical or biophysical applications these amounts could be already sufficient.

DNA template recycling matter was addressed in response to question no. 2 of the reviewer no. 2.

Changes made: We included a paragraph on the scalability in the main text:

*For certain applications, larger amounts of RNA probes might be needed. To test the methodology, we performed PEX on 1 nmol scale with TGK polymerase, **templ_31nt** (Supplementary*

Table S5) and **FAM-RNA-prim_15nt** in two replicates. Natural rNTPs or single modified nucleotide-containing mixtures (**rA^ETP**, rUTP, rCTP, rGTP); (rATP, **rU^{Bio}TP**, rCTP, rGTP); (rATP, rUTP, **rC^{Me}TP**, rGTP) or (rATP, rUTP, rCTP, **rG^{Pent}TP**) were used as representative examples of modifications on each nucleobase. After silica column purification, full-length products confirmed by MS (Supplementary Fig. S241-S260) were quantified against serially diluted synthetic standard by dPAGE (Supplementary Fig. S44). Yields of purified RNA varied from 63% to 78% depending on modification type (Supplementary Table S8).

and to the Discussion section

Therefore, scaling up of PEX in large volume is a viable option, when inexpensive template can be used. Reasonable yields of modified RNA oligonucleotides can be obtained this way, further broadening usability of our method.

We added experimental details concerning scale-up reactions in Supplementary Data file as sections 2.12.8 – 2.12.12 including Supplementary Fig. S44 (dPAGE analysis for yield determination) and Supplementary Tab. S8 (yields overview), Supplementary Fig. S241 – S260 for mass spectra.

Question: 5) The supporting information is very huge. I see that the reactions conditions can be provided in the form of a Table after giving a general procedure for each type of a reaction.

Our response: We thank for the suggestion. However, we consider our layout to be more comprehensible and accessible for the readers and researchers who may wish to reproduce the experiments – we believe that it is impractical to search for parameters scattered all over the Supplementary information. We assume that table format would be rather confusing and less clear representation of experiment setup. Also, every experiment was optimised separately, hence replication of the procedures is more convenient when all instructions are gathered at a single site within the Supplementary Data file.

Changes made: none

REVIEWERS' COMMENTS

Reviewer #1 (Remarks to the Author):

The authors have revised this manuscript to account for the comments of three reviewers (I was Referee# 1). My original review included numerous comments, and the authors have now comprehensively addressed all the points that I raised and certainly markedly improved the quality of the manuscript. The response to comments of the other reviewers were also satisfactory and the authors have added additional and important experiments which clarified some of the issues mentioned in these comments. Particularly, the authors have shown that this method can be applied to longer RNA sequences and that template could be recycled, albeit additional improvements will be required. Hence, publication in Nature Communications is now recommended.

Reviewer #2 (Remarks to the Author):

The authors have addressed all my concerns. I therefore recommend acceptance of the manuscript.

Reviewer #3 (Remarks to the Author):

The authors have revised the manuscript to address all my comments and suggestions. I am satisfied with the revised version. Engineered thermophilic DNA polymerases and strategies to functionalize RNA described in this manuscript are very useful and important. The manuscript is suitable for publication in Nature Communications.